# Diagnostic Significance of Tryptase for Suspected Mast Cell Disorders

**DOI:** 10.3390/diagnostics13243662

**Published:** 2023-12-14

**Authors:** Michiel Beyens, Alessandro Toscano, Didier Ebo, Theo Gülen, Vito Sabato

**Affiliations:** 1Faculty of Medicine and Health Sciences, Department of Immunology, Allergology and Rheumatology, Infla-Med Centre of Excellence, University of Antwerp, 2610 Antwerp, Belgiumalessandro.toscano@uza.be (A.T.); vito.sabato@uza.be (V.S.); 2Department of Immunology, Allergology and Rheumatology, University Hospital Antwerp, 2610 Antwerp, Belgium; 3Department of Immunology and Allergology, AZ Jan Palfijn Gent, 9000 Ghent, Belgium; 4Department of Respiratory Medicine and Allergy, K85, Karolinska University Hospital Huddinge, SE-14186 Stockholm, Sweden; theo.gulen@ki.se; 5Department of Medicine Solna, Division of Immunology and Allergy, Karolinska Institutet, SE-17177 Stockholm, Sweden

**Keywords:** mastocytosis, HaT, MCAS, tryptase

## Abstract

Tryptase has proven to be a very useful and specific marker to demonstrate mast cell activation and degranulation when an acute (i.e., within 4 h after the event) and baseline value (i.e., at least 24 h after the event) are compared and meet the consensus formula (i.e., an increase of 20% + 2). The upper limit of normal determined by the manufacturer is 11.4 ng/mL; however, this boundary has been the subject of debate. According to ECNM and AIM experts, the normal range of baseline tryptase should be 1 to 15 ng/mL. A genetic trait, hereditary alpha tryptasemia, characterized by an increased alpha coding *TPSAB1* copy number is associated with a baseline value above 8 ng/mL. Elevated tryptase can also be found in chronic kidney disease, obesity, and hematological neoplasms. A tryptase > 20 ng/mL serves as a minor criterion to diagnose systemic mastocytosis and an increase in tryptase > 20% + 2 during an acute event is a required criterion in the diagnosis of mast cell activation syndrome. The goal of this review is to demonstrate the (in)significance of tryptase using some clinical vignettes and to provide a practical guide on how to manage and interpret an elevated tryptase level.

## 1. Introduction

Tryptase is the most abundant granule-derived serine protease that is mainly produced by mast cells (MCs) and to a much lesser extent by basophils [1,2,3]. In humans, five isoforms can be found: α-, β-, γ-, δ-, and ε-tryptase [4,5,6]. However, only α- and β-tryptase are clinically relevant and are the most abundant [7]. Resting MCs constitutively secrete monomeric pro-tryptase [8]. In the case of MC degranulation (e.g., during anaphylaxis), MCs release mature tetrameric tryptase. The tryptase assay commercially available from Thermo Fisher^®^ (Waltham, MA, USA) ImmunoCAP measures all of these isoforms in monomeric and tetrameric form (i.e., total serum tryptase) [9].

The predominant indication for tryptase measurement is to document conditions related to systemic MC activation (MCA) (e.g., during anaphylaxis or episodes of mast cell activation syndromes). In this regard, paired sampling is of utmost importance, namely, one during the event (i.e., ideally 30–120 min after the onset of symptoms)—defined as acute serum tryptase (aST)—and one at least 24 h after the event—defined as baseline tryptase (bST). According to the consensus formula, an aST ≥ [(1.2 × bST) + 2], depicts MCA [10,11]. Tryptase can also be a useful marker in suspected primary MC disorders (PMCDs). If bST is more than 20 ng/mL, a minor criterion for the diagnosis of systemic mastocytosis (SM) is met [12,13,14]. Documented MCA based on paired tryptase samples is also a validated and required criterion in mast cell activation syndrome (MCAS). Hereditary alpha-tryptasemia (HαT) is an autosomal dominant genetic trait of an increased copy number of the alpha-coding (of ≥2) *TPSAB1*-gene [15,16], which may account for elevated serum tryptase in the absence of clonal mast cell disorders. The clinical relevance of HαT is still under debate since not everyone with this trait experiences mediator-related symptoms or anaphylaxis [17,18]. However, it seems to be a disease-modifying trait in anaphylaxis and PMCD [19,20,21].

Finally, serum bST levels can be elevated secondary to other numerous non-MC related conditions, the most common being chronic kidney disease (CKD) [22]. The aim of this review is to demonstrate the utility and (in)significance of serum tryptase levels as a screening test in the evaluation of patients with suspected mast cell disorders illustrated with clinical vignettes.

## 2. Clinical Vignettes

### 2.1. Venom-Induced Anaphylaxis with Elevated bST

A 61-year-old woman was stung by a wasp and experienced nausea followed by a syncope 5–10 min later. There was no urticaria or angio-edema. Upon arrival of emergency services, pulseless electrical activity (PEA) was observed. After two cycles of resuscitation and the administration of a total dose of 2 mg adrenaline, a return of spontaneous circulation was obtained. The aST collected upon arrival to the hospital (45 min after the onset of the event) was >200 ng/mL. She had no apparent skin lesions suspicious of mastocytosis. After recovery, one week later, bST was determined and was 20 ng/mL. Specific IgE (sIgE) antibodies for wasp venom (Phadia Thermo Fisher Scientific, Uppsala, Sweden) were positive (2.51 kUA/L reference < 0.10 kUA/L). In this case, the REMA score (explained below) was 3, warranting further investigations for an underlying PMCD. The *KIT D816V* mutation (Bio-Rad Laboratories Inc., Hercules, CA, USA) was detectable in peripheral blood. A bone marrow (BM) examination revealed aberrant MC markers CD2, CD25 and CD30. The pathological investigation showed multiple MC aggregates and spindle-shaped MCs. Taken together, one major and four minor criteria were positive, and thus, SM was diagnosed. Because of the absence of B- and C-findings, indolent SM was the final diagnosis. Treatment with wasp venom-specific immunotherapy was initiated.

### 2.2. Venom-Induced Anaphylaxis with bST within the Normal Range

A 66-year-old woman was brought to the emergency department for treatment of anaphylaxis triggered by a wasp sting. Initially, she felt dizzy and experienced flushing and pruritus, and within a few minutes, she had lost consciousness. On arrival, paramedics administered 0.3 mg of intramuscular adrenaline. In the emergency room, she had difficulty maintaining blood pressure (which dropped to 65 mmHg systolic) and a second dose of 0.3 mg intramuscular adrenaline was administered along with intravenous fluids. Her systolic blood pressure rose to 85 mmHg, and she was observed for 24 h before discharge. Her acute tryptase level was 38 ng/mL during the anaphylactic episode.

After discharge, the patient was referred to our allergy clinic. She had a positive sIgE for wasp venom (0.83 kUA/L), and a diagnosis of wasp venom allergy was confirmed. When further questioned about her medical history, she mentioned a previous reaction to a wasp sting five years ago. On that occasion, she experienced cutaneous (generalized urticaria, flushing) and gastrointestinal symptoms (abdominal discomfort and diarrhea) but did not seek emergency care. The baseline tryptase was 12 ng/mL, and a provisional diagnosis of secondary MCAS was made. There were no typical skin lesions suggestive of mastocytosis, but a *KIT D816V* mutation was detected in the peripheral blood. A BM investigation was performed, and the presence of CD25-expressing aberrant MCs was demonstrated. Moreover, the histopathologic investigation revealed spindle-shaped MCs (>30%) without the presence of dense multifocal MC infiltrates. Thus, she received the diagnosis of indolent systemic mastocytosis by fulfilling three minor criteria of SM. Shortly after diagnosis, wasp venom-specific immunotherapy was started.

### 2.3. Anaphylaxis Mimicking Epilepsy

A 32-year-old man called emergency services because of malaise after an insect sting. Upon arrival, he was found unresponsive in his vehicle. Blood pressure, heart rate, and saturation were normal except for a decreased consciousness. He received 0.5 mg adrenaline IM, after which he regained consciousness, although he was very agitated, disoriented, and had slurred speech. He was unable to execute commands. There were signs of tongue bite and urinary incontinence. Further neurologic investigation showed isochoric and light-reactive pupils and normal reflexes. Because of almost isolated neurological signs and symptoms, a neurogenic disorder was suspected rather than anaphylaxis. In the hospital, a laboratory examination revealed normal cytology, inflammatory markers, liver function tests, electrolytes, creatinine, uremia, and TSH. Toxicology and ethanolemia screening were negative. An ECG was normal, as was a thorax X-ray. A head CT showed no signs of intracranial hemorrhage, no mass effect, and no signs of overpressure. Since meningitis and encephalitis were suspected, a lumbar puncture was performed, and acyclovir and ceftriaxone were started empirically. At the time, the main differential diagnosis was meningitis, encephalitis, withdrawal, or epilepsy. On EEG, no signs of epileptiform activity could be observed. An MRI of the brain revealed no abnormalities or signs of recent ischemia. The next day, neurological recovery was complete except for retrograde amnesia. Because of the initial story of an insect sting, anaphylaxis was reconsidered as a plausible diagnosis. In this respect, aST was measured in the serum sample obtained approximately one hour after the onset of symptoms and was revealed to be elevated (909 ng/mL). The baseline value, taken 24 h after the acute event, was 8.3 ng/mL. This paired sampling confirmed MCA and contributed to the diagnosis of anaphylaxis. An allergy work-up confirmed an IgE-mediated wasp venom allergy. Because of the unusual presentation and very high aST, a work-up for an underlying PMCD was carried out. A *KIT D816V* mutation was found in peripheral blood and bone marrow (BM) (both 0.02%). No other WHO criteria for mastocytosis were found; thus, we concluded monoclonal MCAS (MMAS) as the underlying diagnosis. Venom-specific immunotherapy was started.

### 2.4. Recurrent Anaphylactic Episodes without Elevated bST

A 43-year-old female was referred to our allergy clinic for a second opinion due to unexplained recurrent anaphylaxis that debuted when the patient was 25 years old. She had approximately 50 episodes of MC mediator-related symptoms. Her symptoms predominantly occurred during her menstrual cycles. Five of these episodes were considered to be anaphylaxis and were mostly manifested as a combination of cardiovascular, cutaneous, and gastrointestinal (GI) symptoms. During the remaining episodes, she had isolated cutaneous (i.e., pruritus, flushing) or GI symptoms of abdominal cramps and diarrhea. She had no history of allergies or asthma. Her allergy work-up including a skin prick test (inhalation and food panel), and specific IgE tests were unremarkable. She had a total IgE level of 30 kU/L. Her bST levels were between 2.5 ng/mL and 3.5 ng/mL. A peripheral blood *KIT D816V* mutation analysis was negative, and she had no signs of skin lesions typical of mastocytosis. During the 5 years of follow-up in our clinic, the patient developed two more episodes of anaphylaxis. Altogether, she presented with syncope in five of her seven anaphylaxis episodes. Acute tryptase levels were measured at two of these episodes and were found to be 14 and 12 ng/mL, which were significantly higher than the patient’s baseline levels. We performed a BM biopsy, which showed no signs of clonality. She has been regularly on an H1-blocker Desloratadine^®^ (Sandoz, Basel, Switserland) 5 mg three times daily during the last 3 years and, interestingly, she has not had any anaphylactic episodes since. Finally, the patient obtained the diagnosis of idiopathic MCAS.

### 2.5. Anaphylaxis in a Patient with a Relatively Elevated Baseline Serum Tryptase

A 60-year-old woman suffered from an uncomplicated urinary tract infection and was prescribed nitrofurantoin. About 20 min after intake of the first dose, she experienced pruritus, hives, angioedema of the lips, dyspnea, wheezing, and syncope. In order to maintain proper cardiovascular circulation, adrenaline 0.5 mg was administered twice intramuscularly together with an intravenous administration of adrenaline titrated up to 400 mcg and a continuous drip of noradrenaline. An aST obtained approximately 1 h after the onset of the symptoms was 45.8 ng/mL. Two days after admission to the intensive care unit, she was discharged in good health. A bST one month after the event was 19.1 ng/mL. A BM biopsy was carried out due to the relatively elevated bST combined with the severe reaction despite that the *KIT D816V* mutation was negative in peripheral blood. No aberrant MC markers (CD2, CD25, or CD30) were found on BMMCs. Moreover, the histopathological investigation did not reveal MC clusters or spindle-shaped MCs. An increased copy number of the *TPSAB1*-gene was detected (the alpha- and beta-tryptase copy numbers were three and two, respectively), identifying underlying HαT.

### 2.6. Recurrent Angioedema and Relatively Elevated Baseline Serum Tryptase

A 49-year-old man with recurrent angioedema since 2019 was referred to our allergy clinic. The symptoms started with itching on the palms of the hands or soles of the feet, then, shortly after, swelling up in different parts of the body. In some cases, he even had swelling of the tongue. An allergy work-up was unremarkable. Hereditary and acquired angioedema due to a functional C1-inhibitor deficiency was ruled out, as the complement factor C4 and functional C1-esterase inhibitor analysis were unremarkable. He then started prophylactic treatment with Desloratadine^®^ at three tablets daily, and the number of reactions decreased, although did not completely disappear. His bST level was checked and turned out to be slightly elevated (14 ng/mL). A peripheral blood *KIT D816V* mutation analysis was negative, and he had no signs of cutaneous mastocytosis. He had no history of anaphylaxis. An increased copy number of the *TPSAB1*-gene was detected (with an alpha copy number of two and a beta copy number of three), leading to the diagnosis of HαT. A BM investigation was not performed as the risk of having a clonal mast cell disease was assessed to be insignificant in this case.

### 2.7. Elevated Basal Tryptase without Mediator Related Symptoms

During hemodialysis, a 60-year-old male with a history of end-stage renal failure due to diabetic nephropathy became hypotensive but was resolved quickly after cessation of dialysis and fluid administration. However, since the nephrologists considered anaphylaxis among differential diagnoses, an aST level was obtained. Since his tryptase level, taken 45 min after the hypotensive episode, was 15.3 ng/mL, he was referred to the Department of Allergology for identification of the culprit. Except for a chronic itch without cutaneous signs, the patient had no other mediator-related symptoms. The baseline tryptase taken 24 h after the event was 14.7 ng/mL, which excluded MCA. After the index event, he had multiple sessions of uncomplicated hemodialysis, with exposure to the same excipients. Since the initial event was not suspicious for anaphylaxis and he already was re-exposed to all possible culprits, no further allergic work-up was deemed to be required. Clinical history did not include other elements suspicious for PMCD. A genetic test demonstrated normal copy numbers of the *TPSAB1*-gene; thereby, HαT was ruled out. It was concluded that end-stage renal failure was the cause of the elevated tryptase.

## 3. Discussion

Serum tryptase measures the total concentration of different isoforms of tryptase, mostly α- and β-tryptase. Both can be either in a monomeric (immature) or a tetrameric (mature) form. The monomeric forms are secreted constitutively and can have some variability [23]. During the degranulation of MCs (e.g., anaphylaxis), the mature tetrameric forms of tryptase are released, resulting in an increase in total tryptase [2].

The interpretation of tryptase is dependent on the context. In acute settings, such as suspected anaphylaxis or MC mediator-related symptoms, an aST should ideally be measured from 30 up to 120 min after the start of symptoms since this will correspond with the peak value [24]. However, it may be still measured up to 4 h after a systemic hypersensitivity reaction. Even if the determination of tryptase is not directly available, the serum of the patient should be obtained within this widow. The sample can always be stored or shipped for analysis afterward since tryptase is rather stable [25]. A bST should be taken at least 24 h afterward. Paired analysis of both values allows one to determine if MCA had taken place and should always be checked [26]. Different approaches have been proposed such as a delta tryptase > 3 ng/mL or a rise of 35% in tryptase [27,28]. In perioperative anaphylaxis, the consensus formula showed the best sensitivity and specificity [11]. Overall, the most used and validated approach is the consensus formula.

Using the consensus formula, the aST should have an increase of 20% + 2 ng/mL compared with the bST value [10]. Generally, the severity and magnitude of hypotension in anaphylaxis correlate with the height of tryptase [29,30]. Recently, an aST/bST ratio above 1.685 has been proposed to increase specificity in patients with ISM and/or HαT as these patients can depict a variability in bST without MCAS [23]. However, this requires further validation. For the time being, the consensus formula is still widely used as a criterion and considered as the golden standard [10]. On the other hand, some patients might clearly experience anaphylaxis but not fulfill the criteria for MCA due to a lack of elevation in tryptase. This is especially true for patients with food-induced anaphylaxis [31,32], in which only a 30% increase in tryptase seems to suffice [33]. It is possible that other mediators such as platelet-activating factor (PAF), prostaglandin D2, leukotriene E4, histamine, or other cytokines might be more sensitive in these patients [34,35,36,37]. Note that even when MCA is depicted, no conclusion can be drawn about the mechanism (IgE-mediated or non-IgE-mediated) responsible for MC degranulation [38,39].

An elevated bST can be found in different scenarios that are listed in Table 1. The most common reason is HαT (91%), followed by chronic renal failure (7%) and hematologic malignancies and mastocytosis (1%) [8,19]. HαT is a genetic trade in which there is an increased copy number of the alpha-coding *TPSAB1*-gene. In the majority of cases, it involves a duplication, although more copies (e.g., a quintuplication [40]) have been described. It is the cause of an elevated bST in about 90% of patients and is found in up to 6% of the general population [8,22]. The trait is inherited in an autosomal dominant way. The majority are healthy individuals and will not develop any MC-related conditions [41]. One hypothesis is that the pathological potency of HαT is caused by active heterotetrametric α/β-tryptase and that these are more abundant in patients with a higher α/β ratio [42]. However, more research is needed on this topic. Actually, an increased bST up to 15 ng/mL without any mediator-related symptoms or anaphylaxis is no reason for further evaluation [43]. On the other hand, mastocytosis should be suspected in patients with elevated bST if other common causes of elevated bST are excluded or if the bST exceeds the predicted value based on the number of *TPSAB1* replications. One way to calculate estimated tryptase is to divide the bST by 1 + the extra copy numbers of the alpha-tryptase gene [43]. Another way is by using an online calculator tool (https://bst-calculater.niaid.nih.gov/ (accessed on 13 November 2023)) developed by Chovanek et al. [44].

In some cases, patients with HαT do present with a clinical picture dominated by mediator-related symptoms (i.e., vibratory urticaria, flushing, abdominal cramps, headache, dysautonomia, etc.) [15,16,45]. Some patients might meet the criteria for MCAS [46,47]. However, it is debated whether HαT should be considered as a separate clinical phenotype of MCAS. There is a consensus that HαT is a risk modifier of severe anaphylaxis [21,48,49,50,51,52,53]. However, a recent study did not find a difference in the prevalence of HαT in patients with or without anaphylaxis with underlying SM [54], and more studies with opposing results exist [55].

A bST ≥ 20 ng/mL is a minor criterion for the diagnosis of SM. Importantly, if another hematological neoplasm is present, this criterion is no longer valid, and in the case of HαT, the tryptase level should be adjusted [56]. Moreover, the bST is included in different scoring systems to assess the risk for underlying PMCD in patients presenting with severe anaphylaxis without having typical signs of mastocytosis lesions in the skin. The most commonly used and validated is the REMA-score (Red Española de Mastocitosis (Spanish Network on Mastocytosis)) [57], which was initially developed for *Hymenoptera* venom allergy and later extended to other causes of anaphylaxis. Other scoring systems used to predict clonality are the NICAS score (National Institute for Health and Care Excellence) [58] and Karolinska score [59], which can be used for patients with idiopathic or unprovoked anaphylaxis. These tools use similar parameters such as sex and clinical symptoms. They differ in the cut-off for tryptase and the presence of the *KIT D816V* mutation in peripheral blood, of which the latter is only included in NICAS. Regarding bST, the REMA score includes < 15 (−1 point) and > 25 ng/mL (+2 points); while Karolinska uses < 11.4 ng/mL (−1 point) and >20 ng/mL (+2 points), and finally, NICAS uses only 11.4 ng/mL as cut off [60]. However, tryptase is only one element in the diagnosis. One should be cautious that patients with SM and anaphylaxis seem to have a lower bST in contrast to patients with SM without anaphylaxis [61,62,63,64,65]. Note that the validity of *KIT D816V* detection is highly dependent on the analytic performance characteristics of the applied molecular assay. It is therefore recommended to use a highly sensitive technique such as an allele-specific oligonucleotide quantitative reverse transcriptase polymerase chain reaction (ASO-qPCR) [66] or digital droplet PCR (ddPCR) [67]. Importantly, even with a highly sensitive test, a negative result for the *KIT D816V* mutation in peripheral blood would not exclude underlying PMCD, since this has a low negative predictive value [68,69]. Thus, the diagnosis might be challenging, especially since the clinical presentation can be very heterogeneous [70]. Importantly, tryptase in patients with SM is not correlated with symptom severity [71]. It is however correlated with disease severity since tryptase serves a B-finding if it is above 200 ng/mL (adjusted for the *TPSAB1* status) [56,72].

Non-MC disorders can give rise to bST, such as impaired renal function [73]. Probably, this is due to an elevated SCF since tryptase is not cleared by kidneys [74]. An increase in SCF can induce MC hyperplasia and thus give rise to elevated bST [75]. Tryptase also tends to rise with age [76]. Moreover, obesity can be a cause of elevated bST [77], as well as helminthic infections, hematological malignancies, cardiovascular disease, (nummular) eczema, or rare genetic mutations (e.g., GATA2 or PLAID) [8,78,79]. Even alcohol consumption [80] or tobacco smoking [81] can decrease or elevate bST, respectively. Also, a case of a patient with Gaucher Disease type 1, a lysosomal storage disorder, was reported to have an elevated tryptase level (up to 80 ng/mL) that improved upon initiation of enzyme replacement therapy [82]. Finally, interference with the immunoassay may lead to a false positive result (e.g., by heterophilic antibodies [83]).

Acute tryptase can be used to prove MCA in MCAS [84]. The criteria for diagnosis of MCAS are listed in Table 2. For a complete differential diagnosis of mediator-related symptoms [85] or management of these symptoms [86], the reader is referred elsewhere. The most commonly used biomarker is paired serum tryptase since the consensus formula has been validated. For the diagnosis of MCAS, other biomarkers such as histamine, prostaglandins, leukotrienes, or metabolites can be used. However, one should keep in mind that these metabolites have yet to be validated [34], although an increase of 1.3 in leukotriene E4, 2,3-dinor-11b-prostaglandinF2a, and n-methylhistamine has been proposed as a possible sign of MCA [87]. Additionally, diamine oxidase has been shown to significantly increase during anaphylaxis and has a longer half-life compared with tryptase, making this an interesting marker for further evaluation [88].

## 4. Conclusions

Serum tryptase is an important biomarker in the diagnosis of MC-related diseases. The interpretation strongly depends on the context. In the case of anaphylaxis or mediator-related symptoms, an aST should be obtained 30–120 min after the onset of symptoms, and the result must be paired with a bST obtained at least 24 h after the event. Other clinical clues such as hypotension and the absence of mucocutaneous symptoms may help support the suspicion of an underlying PMCD. In SM, the bST can be elevated; however, a low bST does not exclude diagnosis, especially in patients with a low MC burden. The bST can be elevated in numerous other conditions, the most common being HαT followed by impaired renal function. This should not lead to unnecessary concerns, especially if tryptase is below 15 ng/mL. A proposed algorithm for how to evaluate patients with elevated tryptase without and with mediator-related symptoms is shown in Figure 1A,B, respectively.

Summary box: key messages.

Acute tryptase is determined within 4 h after the onset of symptoms (ideally between 30 and 120 min after the start of symptoms) and should always be paired with a baseline sample at least 24 h later.A baseline tryptase below 8 ng/mL does not exclude primary mast cell disorders.In the case of anaphylaxis or recurrent mediator-related symptoms, a baseline tryptase > 8 ng/mL justifies *TPSAB1*-genotyping.The baseline tryptase can be elevated in various situations. The most common cause is hereditary alpha tryptasemia followed by chronic kidney disease.Hereditary alpha tryptasemia is a genetic trait characterized by an elevated baseline tryptase, which can be asymptomatic but may lead to mediator-related symptoms in some patients and may increase the risk of (severe) anaphylaxis.Tryptase height does not reflect symptom severity in patients with hereditary alpha tryptasemia or systemic mastocytosis.The link between hereditary alpha tryptasemia and systemic mastocytosis remains to be elucidated.

## Figures and Tables

**Figure 1 diagnostics-13-03662-f001:**
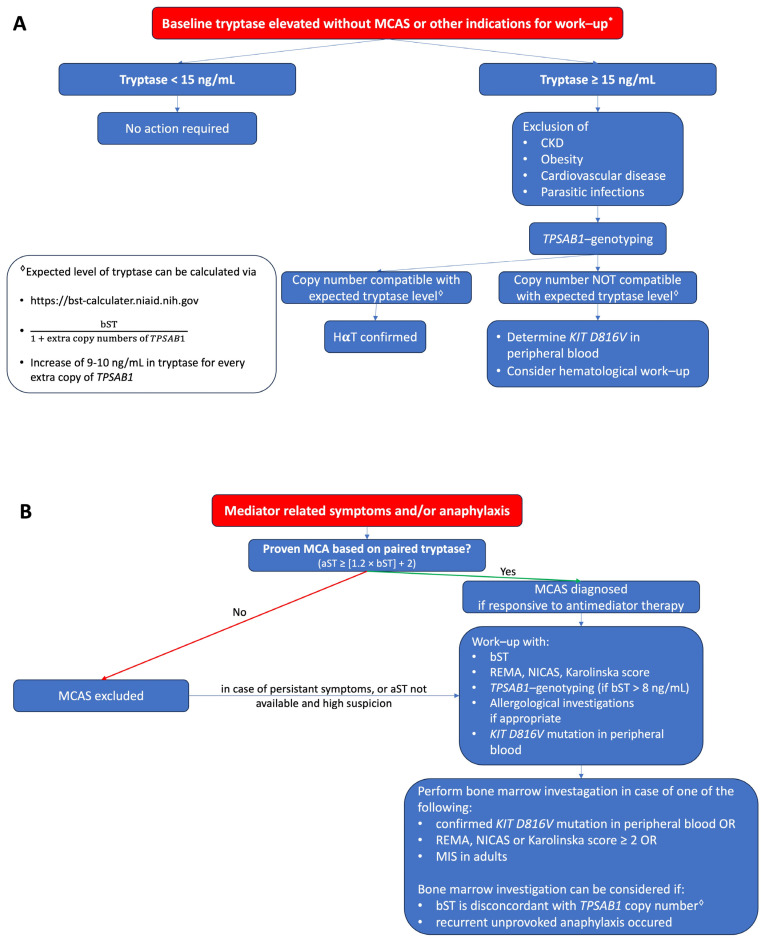
(**A**) Proposed algorithm for patients with elevated bST. (**B**) Proposed algorithm for patients with mediator-related symptoms. MCAS: mast cell activation syndrome; MCA: mast cell activation; CKD: chronic kidney disease; HαT: hereditary alpha tryptasemia; bST: baseline serum tryptase; aST: acute serum tryptase; MIS: mastocytosis in the skin. * Indications for work-up include unexplained osteoporosis or MIS in adults.

**Table 1 diagnostics-13-03662-t001:** Most common etiologies of elevated bST. HαT: hereditary alpha tryptasemia; SM: systemic mastocytosis; SCF: stem cell factor.

HαT
Chronic renal failure
Obesity
Hematologic malignancy (especially myeloid neoplasms)
SM
Chronic parasitic infections (e.g., helminthic infections)
Administration of SCF
Rare genetic mutations (e.g., *GATA2* or *PLCG2*)
Elderly
Cardiovascular disease
False positive (due to interference with the immunoassay)

**Table 2 diagnostics-13-03662-t002:** Required diagnostic criteria for MCAS. Adapted from Valent et al. [13,89]. All three criteria must be fulfilled to establish the diagnosis of MCAS.

Criterion A	Clinical signs of recurrent or severe MCA with involvement of at least two organ systems
Criterion B	Proof of MCA with consensus formula of aST and bST (an increase of 20% + 2 ng/mL) or other biomarkers such as histamine, prostaglandins, leukotrienes, and metabolites
Criterion C	Response to MC-stabilizing drugs, drugs directed against MC mediator production, or drugs inhibiting MC mediator release or inhibiting MC mediator effects

## Data Availability

No new data were created or analyzed in this study. Data sharing is not applicable to this article.

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
