# Peer review of "Diagnostic Significance of Tryptase for Suspected Mast Cell Disorders"

_diagnostics, 2023, doi:10.3390/diagnostics13243662_

Round 1
Reviewer 1 Report
Comments and Suggestions for Authors
The manuscript nicely summarizes the diagnostic use of serum tryptase for patients with suspeted mast cell diseases. The existing literature is comprehensively included the discussion is well balanced. In addition, a number of interesting clinical cases are presented.
I only have some minor points to be considered:
1) Case 2: SM has been diagnosed based on the presence of three minor criteria. A comment on the major criterion is missing. Has this not been fulfilled?
2) The alpha and beta TPSAB1 copy number of patients with HaT should be included in the cases.
3) The clinical validity of KIT D816V testing in peripheral blood heavily relies on the analytical performance characteristics (limit of detection) of the molecular assay applied. The current recommendations of ECNM/AIM on the Standards of genetic testing in the diagnosis of systemic mastocytosis should be briefly discussed.
4) The second paragraph of the discussion describes the necessity of measuring aST during within 2 or 4 hours of an event. From a pratical point of view it would be helpful to add that the blood draw for optaining serum is crucial within this periode even if tryptase measurement is not directly available in the emergency room setting. The serum can then easily be stored/shipped as tryptase is a rather stable analyte.
5) The statement "Importantly, tryptase is patients with SM is not correlated with disease severity (65).", is not true should and should be discussed in more detail. While there is no relevant correlation with severity of symptoms, tryptase is a relevant biomarker for disease burden in SM, a B-finding for definition of SSM and a prognostic marker (e.g. in the ECNM IPSM score).
Comments on the Quality of English LanguageThe manuscript should be thoroughly checked for typos.
Author Response
The manuscript nicely summarizes the diagnostic use of serum tryptase for patients with suspeted mast cell diseases. The existing literature is comprehensively included the discussion is well balanced. In addition, a number of interesting clinical cases are presented.
We thank the reviewer for this kind comment.
I only have some minor points to be considered:
1) Case 2: SM has been diagnosed based on the presence of three minor criteria. A comment on the major criterion is missing. Has this not been fulfilled?
We thank the reviewer for noticing this. The major criterion was indeed not present in this patient. This was added in the manuscript.
2) The alpha and beta TPSAB1 copy number of patients with HaT should be included in the cases.
The TPSAB1 alpha and beta copy number was added in the manuscript. We thank the reviewer for this suggestion. The patient in case 5 had an alpha-tryptase copy number of 3 and a beta-tryptase copy number of 2. The alpha- and beta-tryptase if the patient in case 6 were 2 and 3, respectively.
3) The clinical validity of KIT D816V testing in peripheral blood heavily relies on the analytical performance characteristics (limit of detection) of the molecular assay applied. The current recommendations of ECNM/AIM on the Standards of genetic testing in the diagnosis of systemic mastocytosis should be briefly discussed.
We added this in the paragraph. The paragraph now reads as:
Note that the validity of the KIT D816V detection is highly dependent on the analytic performance characteristics of the applies molecular assay. It is therefore recommended to use a high sensitive technique such as an allele-specific oligonucleotide quantitative reverse transcriptase polymerase chain reaction (ASO-qPCR) (64) or digital droplet PCR (ddPCR) (65). Importantly, even with a highly sensitive test, a negative result of KIT D816V mutation in peripheral blood would not exclude an underlying PMCD, since this has a low negative predictive value (66,67).
4) The second paragraph of the discussion describes the necessity of measuring aST during within 2 or 4 hours of an event. From a pratical point of view it would be helpful to add that the blood draw for optaining serum is crucial within this periode even if tryptase measurement is not directly available in the emergency room setting. The serum can then easily be stored/shipped as tryptase is a rather stable analyte.
We thank the reviewer for this good suggestion. This was added in paragraph, which now reads as:
Even if the determination of tryptase is not directly available, serum of the patient should be obtained within this widow. The sample can always be stored or shipped for analysis afterwards, since tryptase is rather stable (25).
5) The statement "Importantly, tryptase in patients with SM is not correlated with disease severity (65).", is not true should and should be discussed in more detail. While there is no relevant correlation with severity of symptoms, tryptase is a relevant biomarker for disease burden in SM, a B-finding for definition of SSM and a prognostic marker (e.g. in the ECNM IPSM score).
We fully agree with the reviewer. We meant that tryptase is not corelated with symptom severity. This was clarified in the manuscript. These lines now read as:
Importantly, tryptase in patients with SM is not correlated with symptom severity (68). It is however correlated with disease severity, since tryptase serves a B-finding if it is above 200 ng/mL (adjusted for the TPSAB1 status) (56).
Reviewer 2 Report
Comments and Suggestions for Authors
This is an interesting and generally well written review which looks into the complex field of the heterogeneity of mast cell disorders.
It is a very comprehensive work that can help to clarify the sometimes difficult approach to the diagnosis and follow-up of these clinical situations. The approach the authors have used is based on clinical cases, which seems very appropriate and practical for less experienced medical doctors.
All the manuscript should be revised as there are a few spelling mistakes in the text, figures and also in the summary box.
In my opinion, this is a suitable journal to be published so many doctors will have the opportunity to read it.
Author Response
This is an interesting and generally well written review which looks into the complex field of the heterogeneity of mast cell disorders.
It is a very comprehensive work that can help to clarify the sometimes difficult approach to the diagnosis and follow-up of these clinical situations. The approach the authors have used is based on clinical cases, which seems very appropriate and practical for less experienced medical doctors.
All the manuscript should be revised as there are a few spelling mistakes in the text, figures and also in the summary box.
In my opinion, this is a suitable journal to be published so many doctors will have the opportunity to read it.
We thank the reviewer for this kind feedback. The manuscript was fully revised for spelling and grammar.